# Characteristics of Gaseous/Liquid Hydrocarbon Adsorption Based on Numerical Simulation and Experimental Testing

**DOI:** 10.3390/molecules27144590

**Published:** 2022-07-19

**Authors:** Shansi Tian, Zhentao Dong, Bo Liu, Haitao Xue, Valentina Erastova, Min Wang, Haiyang Yan

**Affiliations:** 1Key Laboratory of Continental Shale Hydrocarbon Accumulation and Efficient Development, Ministry of Education, Northeast Petroleum University, Daqing 163318, China; shansi.tian@nepu.edu.cn; 2Institute of Unconventional Oil & Gas, Northeast Petroleum University, Daqing 163318, China; 3School of Geosciences, China University of Petroleum (East China), Qingdao 266580, China; b20010024@s.upc.edu.cn (Z.D.); xueht@upc.edu.cn (H.X.); wangm@upc.edu.cn (M.W.); 4School of Chemistry, University of Edinburgh, David Brewster Road, Edinburgh EH9 3FJ, UK; valentina.erastova@ed.ac.uk; 5No.1 Geological Logging Company, Daqing 163318, China; yanhaiyang_lj@petrochina.com.cn

**Keywords:** occurrence characteristics of shale oil, hydrocarbon vapor adsorption experiment, molecular dynamics simulation, kaolinite mineral

## Abstract

Hydrocarbon vapor adsorption experiments (HVAs) are one of the most prevalent methods used to evaluate the proportion of adsorbed state oil, critical in understanding the recoverable resources of shale oil. HVAs have some limitations, which cannot be directly used to evaluate the proportion of adsorbed state oil. The proportion of adsorbed state oil from HVA is always smaller than that in shale oil reservoirs, which is caused by the difference in adsorption characteristics of liquid and gaseous hydrocarbons. The results of HVA need to be corrected. In this paper, HVA was conducted with kaolinite, an important component of shale. A new method is reported here to evaluate the proportion of adsorbed state oil. Molecular dynamics simulations (MDs) of gaseous/liquid hydrocarbons with the same temperature and pressure as the HVAs were used as a reference to reveal the errors in the HVAs evaluation from the molecular scale. We determine the amount of free state of hydrocarbons by HVAs, and then calculate the proportion of adsorbed state oil by the liquid hydrocarbon MD simulation under the same conditions. The results show that gaseous hydrocarbons adsorptions are monolayer at low relative pressures and bilayer at high relative pressures. The liquid hydrocarbons adsorption is multilayer adsorption. The adsorption capacity of liquid hydrocarbons is over 2.7 times higher than gaseous hydrocarbons. The new method will be more effective and accurate to evaluate the proportion of adsorbed state oil.

## 1. Introduction

Shale oil is crude oil impregnated into the layers of shale rock, silt, and impermeable mudstone [1]. It will, therefore, exist either in an adsorbed or free mobile state [2]. Adsorbed oil is found on the surface of organic matter and minerals, while free oil is mainly found in the center of pores and fractures, and is not affected by any restraints, existing in a complete bulk liquid state [3]. The free oil does not experience the effects of surfaces, remaining highly mobile and can be recoverable through natural elastic energy from fracturing [4]. The low-porosity and low-permeability of shales, combined with the high density and viscosity of the oil, means poor flow, and, as a result, low recovery of oil [5]. Therefore, the effectiveness of shale oil extraction is not dependent on the total amount of shale oil, but rather on the amount of moveable fraction. Theoretically, the maximum amount of movable oil is equivalent of the amount in the free state. Hence, the evaluation and prediction of proportion of adsorbed vs. free oil states is critical. Currently, six methods were used to evaluate this ratio: (i) the oil saturation index method (OSI) [6], (ii) the multi-temperature step rock pyrolysis analysis technique [7], (iii) the nuclear magnetic resonance method (NMR) [8], (iv) the molecular dynamics simulation (MD) [9], (v) the liquid hydrocarbon adsorption experiment (LHA) [10], (vi) the hydrocarbon vapor adsorption experiment (HVA) [11], (vii) sequential extraction [12], and (viii) the X-ray diffraction experiment (XRD) [13].

Javie proposed the OSI based on the exploration experience of Monterey, Barnett, and Eagle ford shale oil [6]. OSI is the ratio of S_1_ (residual hydrocarbon content) to TOC (total organic carbon). The extensive oil test results show a positive correlation between the OSI and the amount of moveable oil [6,14,15,16]. If the ratio is greater than 100, oil crossover is considered to exist [17]. This implies a high potential for shale oil recovery. This evaluation method is widely used in engineering due to its simplicity, but being an empirical-based approach is not accurate enough.

The multi-temperature rock pyrolysis analysis technique is based on the difference in the energies of thermal evaporation between absorbed and free oil [7]. The free state oil in fractures and macropores can be easily thermally released compared to adsorbed state oil on the pore surface [18]. The results reflect the effects of scale reservoir space, molecular mass, and polarity on oil mobility, but do not directly inform on the amount of oil in each state [19].

In recent years, with the increased availability of computational resources and method developments, there has been a rise in the applications of molecular modelling to the shale oil occurrence research [4,5,12,14,15,16,18]. MD, being one of such techniques, allows to effectively simulate the interactions between oil molecules, surrounding minerals, and organic compounds. From here, the distribution of oil molecules on a range of surfaces and reservoir will aid the understanding of the mechanisms behind the oil mobilization. From such distributions, one can also compute the proportion of the adsorbed vs. free oil [3,4,20]. However, such simulations are often at an atom-level resolution, the studies remain on 10 s of nm scales, involving only a set of molecules and conditions predefined by the user. Therefore, the modelled system may not be representative of a bigger and more complex realistic system. It is, therefore, essential that the model is validated against the adsorption experiments similar to the simulated system.

To this end, LHA are appropriate to test the accuracy of MD [21,22]. In the LHA, the adsorbate is dissolved in an organic solvent. After a period of time, the adsorption capacity is evaluated by establishing the relationship between the adsorption capacity and the equilibrium time. However, the effect of the organic solvent on the adsorption amount is significant and cannot be removed [10,22]. Therefore, LHA cannot be combined with molecular dynamics simulations.

On the other hand, HVA compensates for the deficiencies of LHA, making them an ideal choice for the validation of the MD [2,11,23,24]. In the HVA, molecules from the gas adsorbed onto the surface of the porous material in a layer-by-layer manner, depleting their density in the gas, until the equilibrium is reached. The equilibrium will be reached when the gas pressure, P, is equivalent to the saturated vapor pressure, P_0_. Therefore, as a relative pressure (P/P_0_) increases, the adsorbed volume of molecules also increases. As the pressure is increased further molecules condense onto the adsorbed ones, forming the capillary condense, which fills micropores to macropores sequentially. Eventually, the pores are filled with adsorbed and capillary condensed fluids, analogous to the adsorbed and free state fluids within the reservoir-saturated oil pore space [2].

However, there are still questions to be answered about HVA. The proportion of adsorbed state oil from HVA is always smaller than that in shale oil reservoirs. Shale oil adsorption experiments are composed of liquid hydrocarbons, while in the HVA gaseous hydrocarbons are used. Currently, no relevant studies prove that the number of adsorption layers, adsorption thickness, adsorption density, and adsorption amount per unit area is directly comparable for these experiments. Therefore, the interpretation of HVA should be revisited with the aid of MD, bringing atom-level details into the process [2]. In this work, we combine the HVA with MD of gaseous hydrocarbons.

Although the interaction of hydrocarbon molecules with pores and surfaces coated with kerogen-like materials has been more extensively studied, conceptually shale consists of two parts: organic matter (kerogen) and inorganic matter (minerals). The inorganic part of shale mainly contains quartz, calcite, feldspar, and clay minerals. Each kind of mineral makes up a certain volume fraction of a lacustrine/marine shale and plays an important role in shale systems through presenting intra- and interparticle pore networks that may hold hydrocarbons. Studying the interface of inorganic pores with oil is challenging. Kaolinite often forms surface coatings in the inorganic pores of shale reservoirs, as well as forming pore filling aggregates and presents interparticle and intraparticle pore surfaces [20]. Kaolinite has unique physicochemical properties. Unlike smectite clays, it is non-swelling, but has both a hydrophobic siloxane surface and hydrophilic aluminol surface. Studying the adsorption of hydrocarbons on kaolinite surfaces will further assist in the understanding of the adsorption characteristics on both polar and non-polar surfaces. Therefore, kaolinite was chosen as the object of study.

This paper seeks to evaluate the proportion of adsorbed state shale oil by HVA and MD. In Section 2.1, the information of samples and the process of HVA testing were introduced. In Section 2.2, the flows of MD simulation of gas/liquid hydrocarbons are shown. In Section 3.1, the adsorption characteristics of gaseous and liquid hydrocarbons are compared. In Section 3.2, the results of HVA experiments are interpreted at the molecular level. A correction method for HVA is proposed for the evaluation of the proportion of adsorbed state shale oil.

## 2. Materials and Methods

### 2.1. Experimental

#### 2.1.1. Materials

N-pentane (≥95.0 wt% or A.R. grade or whatever if known, supplied by Sinopharm Chemical Reagent Co., Ltd., Beijing, China) was used as the adsorbate in the HVA. Table 1 shows the specific physical parameters.

The kaolinite clay (KGa-1 b, low-defect, supplied by Clay Minerals Society) was used as an adsorbent in the experiments. The KGA-1 b kaolinite clay structure is provided as (Mg_.02_ Ca_.01_ Na_.01_ K_.01_)[Al_3.86_ Fe(III)_.02_ Mn_tr_ Ti_.11_][Si_3.83_Al_.17_]O_10_(OH)_8_ and supplied as 96% kaolinite with a trace dickite [26]. The reported cation exchange capacity is 3.0. The reported cation exchange capacity is 3.0 ± 0.1 meq/100g [27] and kaolinites display a moderate hydrophilicity (surface electron-donicity 30–35 mJ/m^2^) [28].

The kaolinite samples were milled into powder particles through 40–60 mesh (250–425 µm) by an agate mortar. The low-temperature nitrogen adsorption/desorption (LT-N_2_ A/D) were measured over relative pressures ranging from approximately 10^−5^ to 0.995 using an Autosorb-iQ-Station-1 instrument at 77 K to obtain the pore size distributions and specific surface areas of the kaolinite samples [29,30]. Figure 1 shows the pore volume distribution determined by Barrett–Joyner–Halenda (BJH) method [31] (with total pore volume of 0.136 cm^3^/g) and surface area distribution determined by BJH with a peak of 12.27 m^2^/g, which is in an agreement with the reported 13.1 m^2^/g determined by BET [32].

#### 2.1.2. Hydrocarbon Vapor Adsorption Methodology

The hydrocarbon vapor adsorption experiments were performed with the 3H-2000 PW multi-station weight method vapor adsorption instrument [33,34] (from Bayside Instrument Technology Co., Ltd., Beijing, China). The instrument includes the evacuation system, constant temperature system, measurement chamber, liquid distillation, and purification system (see schematic on Figure 2). 

*High purity sorbent extraction.* We connected tube A with n-pentane and tube B to the liquid distillation purification system. We then connected tube A to the evacuation system, and kept heating the distillation tube A to remove the low boiling point impurities. After removing the low boiling point impurities, we connected tube A to tube B. We heated tube A to evaporate the adsorbate to a vapor state and then condensed it in tube B under a liquid nitrogen environment. Then, the remaining small amount of adsorbate reagent (mostly high boiling point impurities) of distillation tube A was replaced with a clean distillation tube A. Finally, by repeated distillation (A and B), a high purity adsorbate was obtained from distillation tube A.

*Remove gas impurities from the sample and the device.* After loading the sample into the sample tube, the sample tube was heated to 110 °C. The adsorbates such as air, water, and hydrocarbons that were initially present in the pores of the device and the sample were removed by the evacuation device.

*Buoyancy correction by helium method.* At 313 K, helium of different pressures is passed into the test chamber. Assuming helium to be a non-adsorbing gas, the observed change in the sample mass as a function of pressure, *P*, can be attributed solely to the buoyancy factor [35]:(1)$$V_{c}=-\left(\frac{∆m_{n}}{∆P}\right)\left(\frac{RT}{MZ}\right)$$
(2)$$m_{a}^{ex}=∆m_{n}+∆\rho_{g\;}V_{C}$$
where *V_c_* is collective volume, △*m_m_* is mass of measured, *R* is the gas constant, *M* is molar mass of helium, *Z* is the compressibility factor, △*ρ*_g_ is the density change of helium, and *m_a_^ex^* is excess adsorption amount.

*The adsorption isotherm was measured.* The temperature of the experiment was set to 313 K by the thermostat control system. Then, the pressure was gradually increased, and the weight change of the sample before and after adsorption under a certain relative pressure P/P_0_ was weighed by a microbalance. The isothermal adsorption curves were obtained by recording the data.

#### 2.1.3. Data Analysis

The saturation vapor pressure (P_0_) of n-pentane is calculated by the Clausius–Clapeyron equation. The Clausius–Clapeyron equation enables the determination of the vapor pressure of a liquid at different temperatures if the enthalpy of vaporization and vapor pressure at a specific temperature is known. For this purpose, the linear equation can be expressed in a two-point format. If the vapor pressure is *P*_1_ at temperature *T*_1_, and *P*_2_ at temperature *T*_2_, the corresponding linear equation is:(3)$$\ln\frac{P_{2}}{P_{1}}=-\frac{L}{R}\times \left(\frac{1}{T_{2}}-\frac{1}{T_{1}}\right)$$
where *L* is specific latent heat, also known as enthalpy of vaporization, and *R* is the gas constant. 

### 2.2. Molecular Simulation Details

#### 2.2.1. Molecular Structures

The kaolinite clay has a very small number of isomorphic substitutions, therefore, for this study, the unit cell was assumed to be Al_2_Si_2_O_5_(OH)_4_ (Figure 3a). The atomic positions of the initial crystals were taken from the American Mineralogist Crystal Structure Database [36], with the dimension of 0.5148 nm × 0.892 nm × 0.63 nm, and angles of 90°, 100°, and 90° [37]. The kaolinite mineral slab was composed of three stacked layers, each periodic in the *xy*-plane and comprising of 12 × 7 unit cells. The total mineral model, therefore, is made up of 252 (12 × 7 × 5) unit cells, and creating a slab with dimensions of 6.178 nm × 6.244 nm × 1.89 nm (Figure 3b). In the simulation setup, the kaolinite slab is placed in the region 0 < *z* < 1.9 nm, which fluctuates slightly during the simulation. The pore region above the mineral surface is set to 8 nm, resulting in the simulation box of 6.178 nm × 6.244 × 9.9 nm. This slit-pore setup is representative of the nanoscale pores identified in shale reservoirs, where pores at 8~20 nm occupy a considerable (21.5~47.9%) proportion [20,38].

As in the experiment, n-pentane (n-C_5_H_12_) was used to perform both gaseous and liquid hydrocarbon adsorption simulations. In the gaseous hydrocarbon adsorption simulation, the model was loaded with 10, 25, 50, 100, and 200 molecules of n-pentane at a fixed pore volume, in order to represent different relative pressures of the system. For the liquid hydrocarbon adsorption simulations, 2000 molecules of n-pentane were loaded into the pore, at which point the volume was then allowed to maintain a pressure of 100 bar (the calculation process is as shown in [39]). The density of 2000 n-pentane molecules in the box is the same as the density of n-pentane at 100 bar. The Packmol [40] program was used to insert n-pentane molecules into the kaolinite pore model. The initial model is shown in Figure 3.

#### 2.2.2. Force Field Parameters 

Kaolinite clay was modelled with ClayFF [41] force field and n-pentate with the Charmm36 force field [42], assigned with Cgenff [43]. Both force fields use the Lorentz–Berthelot mixing rule for the non-bonded interactions. Previous studies have confirmed the reliability of the use of these two force fields together [4], and that the results are not only consistent with the ab initio molecular simulations, but also with the results of X-ray diffraction experiments [44].

#### 2.2.3. Simulation Protocol 

The simulations were carried out using the GROMACS 4.6.7. engine [45]. First, every setup system was energy minimized using the steepest descent algorithm, with the convergence criterion being the maximum force on the atom being less than 100 kJ/mol/nm. All simulations were performed using real-space particle mesh Ewald (PME) electrostatics, and van der Waals cut-off was set to 1.4 nm. The simulation step size was 1 fs, and all H-bonds were restrained. The temperature of the MD simulations was kept at an experimental 313 K, using a velocity-rescale thermostat with the temperature coupling constant of 0.1 ps.

The gas adsorption simulations were carried out in the canonical (NVT) ensemble to ensure that the volume of the interlayer is constant. The systems were equilibrated for 0.5 ns, then a production run of 30 ns was performed.

Liquid adsorption simulations were performed in the isothermal-isobaric (NPT) ensemble. The pressure of 100 bar was applied semi-isotopically, ensuring the decoupling the *xy*-plane from the pore space in the *z*-direction. The pressure of 100 bar was applied semi-isotopically, ensuring the decoupling of the *xy*-plane from the pore space in the *z*-direction. The pressure was coupled at 1ps using the Berendsen barostat. After 0.5 ns equilibration, a production run of 30 ns was performed.

#### 2.2.4. Data Analysis

The trajectory from the last 10 ns of the production run was used for analysis.

*Linear mass density*, *ρ*(*z*), was computed using GROMACS tools [46]. The density is defined as a mass, *m*, per volume. In the case of a linear density, it is sampled with 1000 equivalent windows along the *z*-coordinate. Each window having a volume, *V_window_*, which is determined by the length of the window, Δ*Z*, and the area of the surface, *A_surf_*, in *xy*-direction. *A_surf_* can be calculated by the open source wave function software Mut and the molecular visualization software VMD jointly [47,48].
(4)$$\rho \left(Z\right)=\frac{m}{V_{window}}=\frac{m}{A_{surf}\;∆Z}$$

The density profile is calculated as an average for the 10 ns trajectory. The linear density of the pentane system is shown on Figure 4.

The adsorption capacity, *C_Ads_*, is defined as adsorbed mass, *m_ads_*, per unit area, *A_surf_*, here kaolinite surface.
(5)$$C_{Ads}=\frac{m_{Ads}}{A_{surf}}$$

Here, the adsorbed mass is calculated from linear mass density of adsorbed layers, *ρ_Ads_*, spanning the space of the length, *L*, between *z* = *L*1 and z = *L*2:(6)$$m_{Ads}=\int_{z=L1}^{z=L2}A_{surf}\rho {\left(z\right)}_{Ads}\;dz$$

The free phase density, *ρ_gas_*, is determined as an average of the linear density on the gaseous phase, *G*, spanning from *z* = *G*1 to *z* = *G*2.
(7)$$\rho_{gas}=\int_{z=G1}^{z=G2}\rho \left(z\right)\;dz$$

The adsorption density profile of n-pentane in the gaseous adsorption model can be obtained from Equations (3) and (4) (Figure 4), and the average free phase density can be determined. The relative pressure corresponding to the free phase density of n-pentane at 313 K can be found in the NIST database (Figure 5). From this, the relative pressures can be calculated for loading various molecular number models.

## 3. Result and Discussion

### 3.1. Comparison of Adsorption Characteristics of Gaseous and Liquid Hydrocarbons

The adsorption densities of n-pentane systems can be obtained from the partial density profiles, as shown in Figure 6. Table 2 gives the obtained values for the densities of the adsorbed layers and the bulk and the associated relative pressure of the systems.

It can be seen in Figure 6 that gaseous hydrocarbon will first adsorb as a monolayer on both surfaces, and with the increasing relative pressure, the peak density and adsorption thickness increase. It is worth noting that the peak density of the adsorption layer can be even smaller than liquid hydrocarbon when the relative pressure is lower than 0.46. The characteristics of the first adsorption layer of gaseous hydrocarbons on the silicon–oxygen and aluminum–oxygen surfaces are almost identical. As the relative pressure increases from 0.27 to 0.97, the peak adsorption density of the surface increases from 100.79 to 1128.88 kg/m^3^, and the adsorption thickness increases from 0.32 to 0.72 nm. When the relative pressure is 0.97, the second adsorption layer forms. The second adsorption layer has the same adsorption thickness as the first adsorption layer, yet the adsorption density is much lower than the first adsorption layer.

The partial density curve (green line) of liquid n-pentane shows the characteristics of multilayer adsorption (Figure 6). The density of the first adsorption layer in the liquid is higher than that in the gas systems. The adsorption peaks on the silicate and aluminol surfaces differ in density. The light red region between the curve of liquid n-pentane adsorption density (green line) and the curve of relative pressure of 0.9732 gas n-pentane adsorption density is the larger part of liquid adsorption than that of gas adsorption.

Table 3 summarizes the adsorption capacity calculated directly from the partial densities and derived for the relative pressures of n-pentane. The adsorption capacity of gaseous n-pentane increases with the increase in relative pressure, and the maximum is 0.305 mg/m^2^. In comparison, the liquid is 0.80 mg/m^2^ on the aluminol surface and 0.82 mg/m^2^ on the silicate. The adsorption capacity of liquid is 2.7 times of gaseous at 313 K. It can be seen that the gaseous hydrocarbon adsorption (P/P_0_ ≈ 1) obtained by the vapor method experiment is hardly representative of the liquid state adsorption of shale oil in the geological situation (P/P_0_ = 1). The final results need to be corrected if the shale oil adsorption is evaluated using the vapor method experiment.

A model for calculating gaseous hydrocarbon adsorption is presented (P/P_0_ = 0.1–1), based on the results of molecular dynamics calculations. The monolayer adsorption was performed at a relative pressure less than 0.8. The adsorption thickness and adsorption density have a linear relationship with the relative pressure simultaneously, so a quadratic polynomial was used to fit the relationship between the relative pressure and the adsorption capacity. It is double-layer adsorption at relative pressures 0.97. When the relative pressure increases, the adsorption thickness no longer changes, and only the adsorption density rises linearly. Therefore, a linear equation was used to fit the relationship between relative pressure and adsorption capacity at this stage. According to the fitting results in Figure 7, the adsorption amount for any relative pressure can be calculated from the model.

### 3.2. Comparison of Experimental and Simulated Gaseous Hydrocarbon Adsorption Characteristics

The results of the HVA are shown in Figure 8. HVA adsorption consists of adsorbed and condensed hydrocarbons. Thus it refers to total hydrocarbons *Q_t_*. Under 0.55 P/P_0_, the adsorption amount of HVA grows slowly, varies 3.8 to 4.2 mg/g. Between 0.55 and 0.81 P/P_0_ the adsorption amount of HVA increased exponentially from 4.2 to 11.7 mg/g, which may be due to the massive formation of capillary condensed hydrocarbons. Although the MD adsorption showed a relatively slow linear increase from 0.02 to 1.61 mg/g, under 0.81 P/P_0_. In this work, the HVA interpretation model established by Li [2] was used, and the results of gaseous hydrocarbon MD were taken as critical parameters to calculate the adsorption/free oil proportion.

Figure 9 illustrates the occurrence characteristics of n-pentane in different pores (A, B, C) of kaolinite. Firstly, the hydrocarbon forms adsorption layers on pore surfaces. The pores of zone A (pore diameter r < d_a_) are entirely filled with adsorbed hydrocarbons. According to the Kelvin equation, n-pentane coalesces in the B zone (2 h < r < dk) and becomes a liquid hydrocarbon on the surface of the adsorbed hydrocarbon. Eventually, the pores in the B zone are entirely filled with adsorbed hydrocarbons and condensed hydrocarbons. In contrast, n-pentane does not condense in the C region (dk < r< dmax) and remains in the gaseous state at the adsorbed hydrocarbon surface.
(8)$$d_{a}=2h\;$$
(9)$$d_{k}=2\left(nh+r_{k}\right)$$
(10)$$r_{k}=-\frac{2\sigma V_{L}}{ln\left(P/P_{0}\right)RT}$$
where *h* is the adsorption thickness; *n* is the number of adsorption layers; *σ* is the surface tension, 24.3 dyn/cm; *V_L_* is the molar volume, m^3^/mol; *T* is the temperature, 313 K; *R* is the gas constant, 8.314 P·m^3^/mol/K.

The model for calculating the proportion of adsorbed state oil *r* was established by the amount of adsorbed hydrocarbons *Q_a_* and condensed hydrocarbons *Q_c_*.
(11)$$r=\frac{Q_{a}}{Q_{a}+Q_{c}}$$

*Q_a_* can be calculated from Equation (12). Due to the influence of pore morphology, n-pentane cannot achieve complete theoretical adsorption, and a coefficient *k* is proposed to correct this inaccuracy.
(12)$$Q_{a}=kC_{ave.}S$$

*Q_c_* is calculated by Equation (13). The condensed hydrocarbon volume *V_con_.* is the difference between the effective volume *βV_B_* and the adsorbed hydrocarbon volume *V_ab_.* Equation (14). Owing to the non-homogeneity of the kaolinite surface, only part of the volume in the B zone effectively contributes to hydrocarbon occurrence. The effective pore volume *βV_B_* is used to represent this volume. *V_B_* is the B-zone pore volume, *V(r)* is the pore volume distribution curve, as shown in Equation (16). The adsorbed hydrocarbon volume *V_ab_.* can be obtained from Equation (15). Where *S_B_* is the B-zone specific surface area, *S(r)* is the specific surface area distribution curve, as shown in Equation (17).
(13)$$Q_{c}=V_{con.}+\rho_{con.}$$
(14)$$V_{con.}=\beta V_{B}-V_{ab}$$
(15)$$V_{ab}=kS_{B}h$$
(16)$$V_{B}=\int_{d_{a}}^{d_{k}}V\left(r\right)dr$$
(17)$$S_{B}=\int_{d_{a}}^{d_{k}}S\left(r\right)dr$$

The optimal *k* and *β* of the model were calculated using the MATLAB Optimization Toolbox (*k* = 0.9, *β* = 0.75). The *Q_t_* calculated by the model is close to the experimental results for relative pressures between 0.55 and 0.80, which justifies the application of the model (Figure 10). Additionally, there is an error between the model and experimental results at a relative pressure of 0.24. This implies that the model may be inapplicable at lower relative pressures.

The adsorption ratios derived from the HVA experiments and MD simulations under 0.81 P/P_0_ are shown in Figure 11. Moreover, according to the established HVA model, the adsorption ratio at the relative pressure close to 1 (0.97 P/P_0_) was calculated. Below 0.67 P/P_0_, the n-pentane adsorption ratio increases from 0.13 to 0.24, which is caused by a greater growth rate of adsorption than condensation of hydrocarbons. Above 0.67 P/P_0_, the adsorption ratio decreases rapidly from 0.24 to 0.06, and here the condensation hydrocarbons are generated much faster than the adsorbed hydrocarbons. The adsorption ratios at 0.97 P/P_0_ are much lower than the previous understanding of adsorbed hydrocarbons in shale oil reservoirs. The differences in the adsorption of gaseous and liquid hydrocarbons have already been mentioned. The adsorption capacity of liquid hydrocarbons is over 2.7 times higher than gaseous hydrocarbons. Therefore, the liquid hydrocarbons adsorption amount should be taken into the HVA model instead of gaseous hydrocarbons adsorption amount. After correction, the adsorption ratio is 0.22, which is consistent with the understanding of shale oil in situ development.

Using molecular modeling allowed us to gain a detailed insight into the mechanism of gaseous adsorption of the linear alkanes on the kaolinite mineral surfaces. The values derived from the simulations, nevertheless, were not in agreement with experimentally measured ones. The system presented here did not incorporate complex behaviors of geological systems studied experimentally, in particular: inclusion of edges, preferential exposed surfaces, pH sensitivity of the clay, long-range surface tension effects, impurities, contaminations of the geological sample, and capillary effects in the experimental system.

To create a comprehensive picture, stepwise incorporation of levels of complexity should be completed. Therefore, to further understand the adsorption of oil components on clays, and to link theory and the experiment, the limitations in understanding both methods should be addressed. We envision a joint experimental–computational work to identify and inform about the influence of ratios of the edges to surfaces in natural clays, and the prevalence of exposed aluminol or siloxane surfaces, and the effect of the pH and any impurities.

## 4. Conclusions

(1) The hydrocarbon vapor adsorption experiment (HVA) cannot directly assess the proportion of adsorbed state oil due to the difference in adsorption of gaseous hydrocarbons at 0.97 P/P_0_ and liquid hydrocarbons: thickness (2.1 to 3.9 nm), adsorption density (1125 to 1444 kg/m^3^), and adsorption capacity per unit area (0.3 to 0.81 mg/m^2^).

(2) A new method is developed to evaluate the proportion of adsorbed state oil on shale, which is validated by n-pentane adsorption on kaolinite. Shale oil adsorption ratio under 0.8 P/P_0_ and 313 K is 0.05, and is obviously lower than the that in shale oil reservoirs. After correction, the adsorption ratio is 0.22, which is consistent with the understanding of shale oil in situ development.

(3) The adsorption characteristics of unsaturated n-pentane are summarized. Below 0.67 P/P_0_, the n-pentane adsorption ratio increases from 0.13 to 0.24, which is caused by a greater growth rate of adsorption than condensation hydrocarbons. Above 0.67 P/P_0_, the adsorption ratio decreases rapidly from 0.24 to 0.06, and here the condensation hydrocarbons are generated much faster than the adsorbed hydrocarbons.

## Figures and Tables

**Figure 1 molecules-27-04590-f001:**
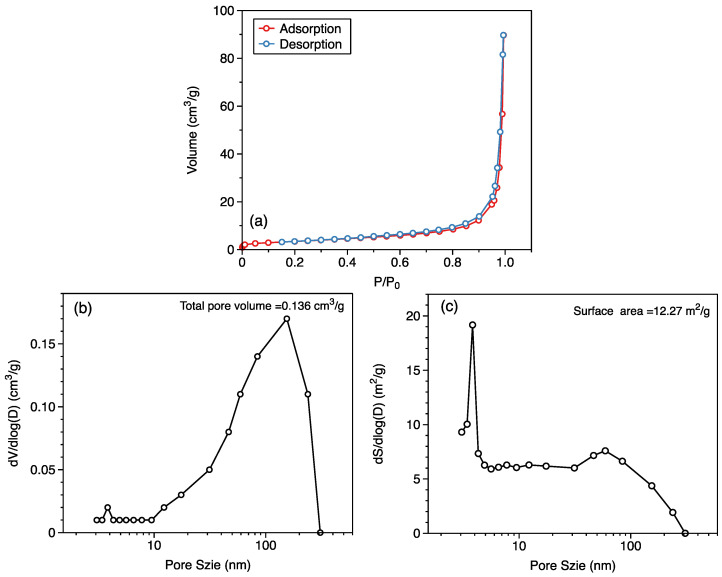
Adsorption/desorption curves (**a**), pore volume (**b**), and pore surface (**c**) distribution obtained by LT-N2 A/D for kaolinite KGA-1b clay mineral.

**Figure 2 molecules-27-04590-f002:**
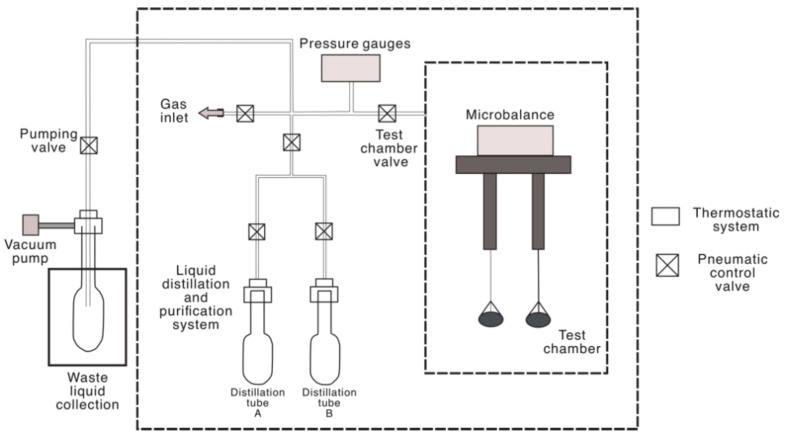
The 3H-2000 PW multi-station weight method vapor adsorption instrument.

**Figure 3 molecules-27-04590-f003:**
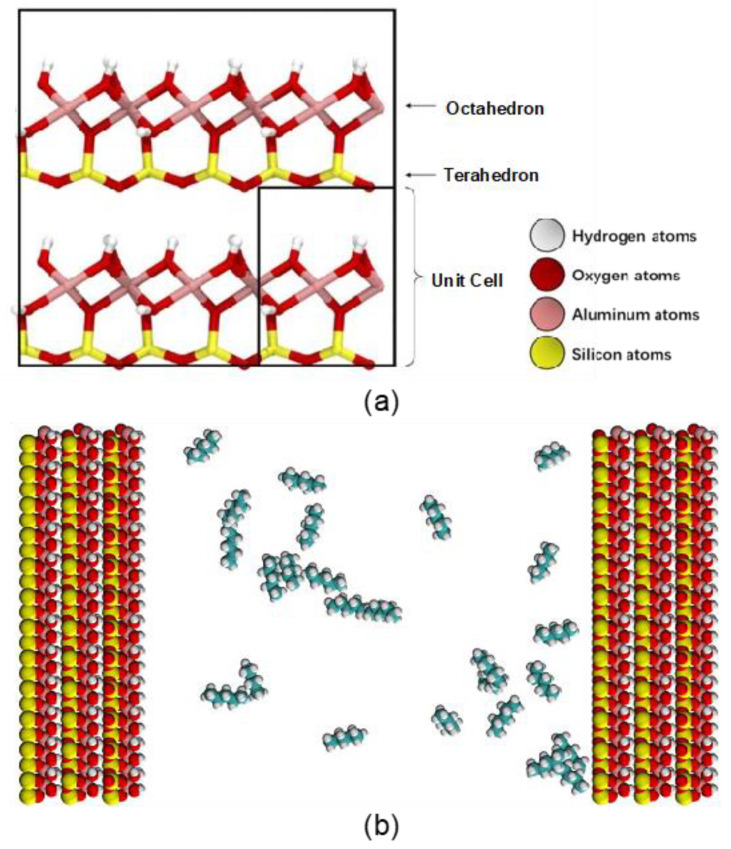
(**a**) A unit cell of kaolinite and (**b**) initial configuration of the kaolinite pore filled with 25 pentane molecules. Kaolinite, with aluminol surface on the left side and silicate surface on the right. The colors are stated here: white (hydrogen atoms), red (oxygen atoms), pink (aluminum atoms), yellow (silicon atoms), and green (carbon atoms).

**Figure 4 molecules-27-04590-f004:**
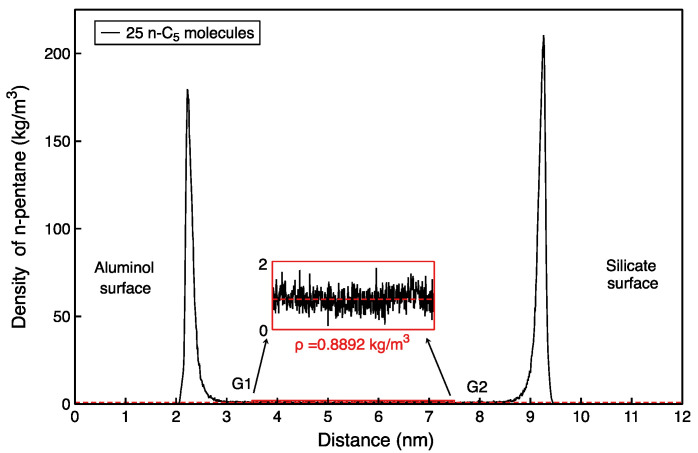
Density distribution curve of 25 n-pentane molecules with free phase hydrocarbon density of 0.8892 kg/m^3^.

**Figure 5 molecules-27-04590-f005:**
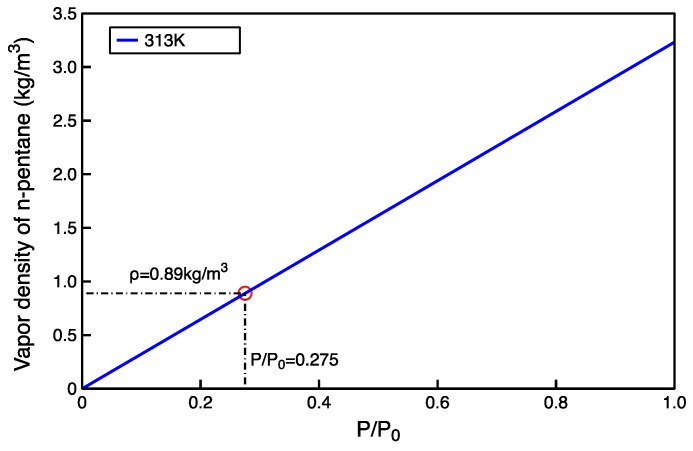
Density map of n-pentane at different relative pressures (according to NIST [44]).

**Figure 6 molecules-27-04590-f006:**
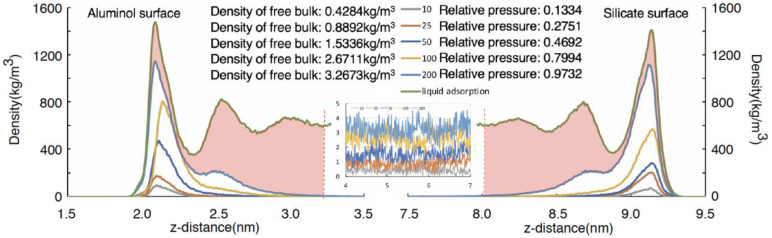
Adsorption density curve of liquid n-pentane and gaseous n-pentane at different relative pressures.

**Figure 7 molecules-27-04590-f007:**
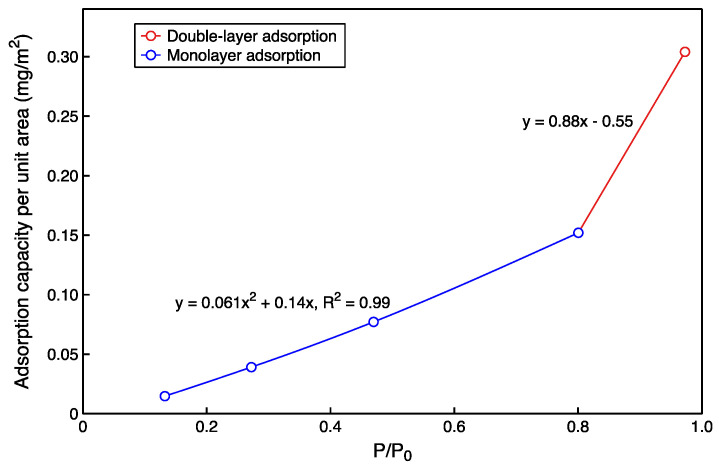
The mathematical model for predicting the amount of hydrocarbons adsorbed per unit area of kaolinite at 313 K.

**Figure 8 molecules-27-04590-f008:**
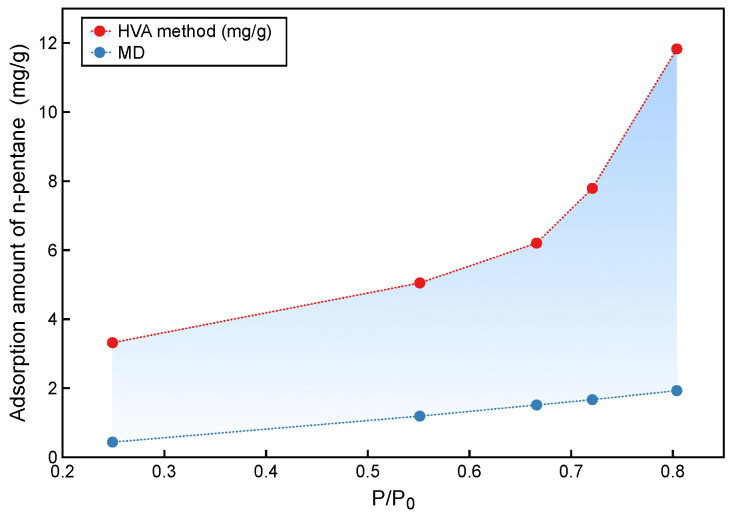
Variation of n-pentane adsorption with relative pressure obtained from HVA at 313 K.

**Figure 9 molecules-27-04590-f009:**
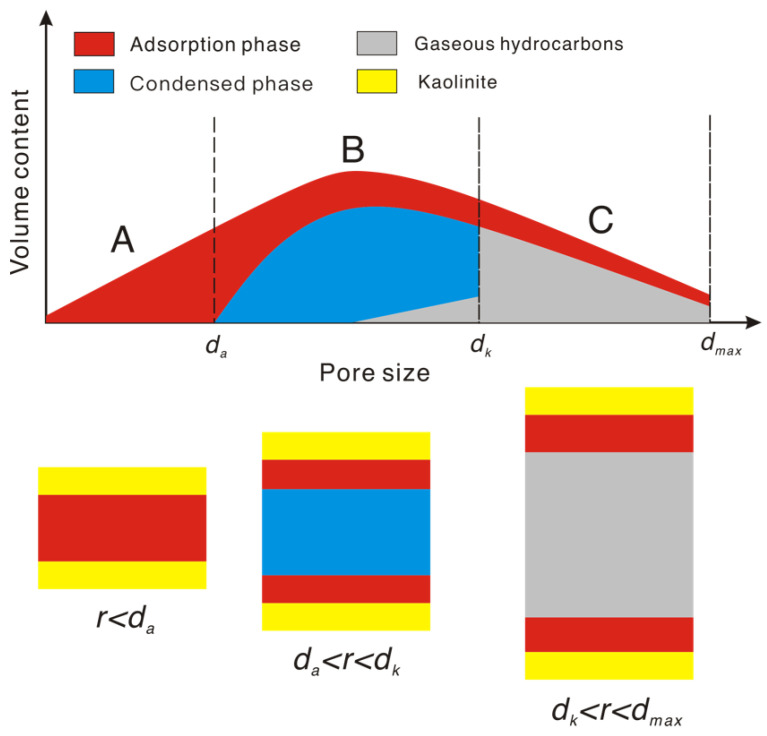
Occurrence of n-pentane in various pores of kaolinite by hydrocarbon vapor adsorption. *d_a_*, *d_k_*, and *d_max_* divide the pores of kaolinite into three zones (A, B, and C). *d_a_* is twice the thickness of the adsorption layer. *d_k_* determines the interval in which vapor coalescence occurs, which can be calculated from the Kelvin radius *r_k_*. *d_max_* is the maximum pore size of kaolinite.

**Figure 10 molecules-27-04590-f010:**
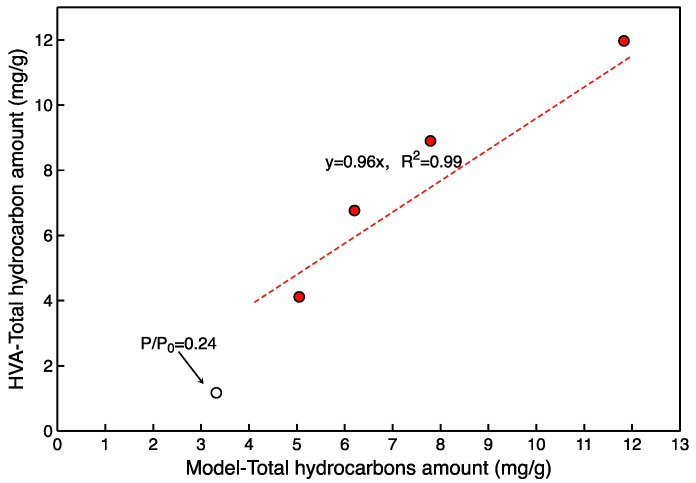
Comparison of model calculation of total hydrocarbons with the results of experiments (*k* = 0.9, *β* = 0.75).

**Figure 11 molecules-27-04590-f011:**
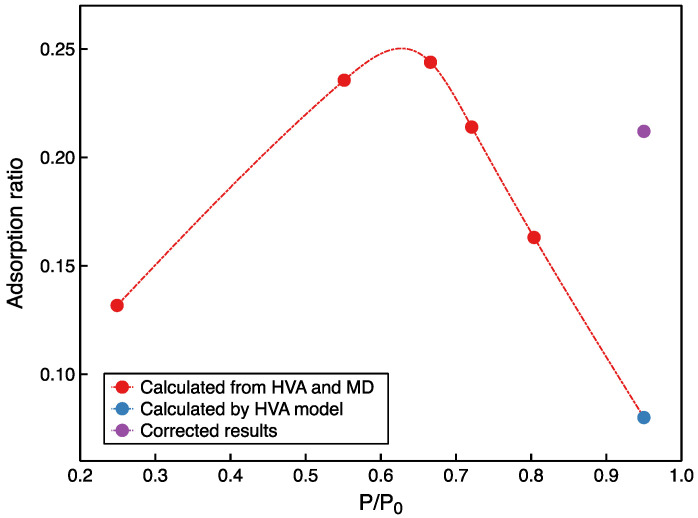
The adsorbed hydrocarbon proportion at different relative pressures in kaolinite at 313 K.

**Table 1 molecules-27-04590-t001:** Physical properties of n-pentane saturated vapor pressure (P_0_), molar mass (M), molar volume (V_L_), and density (ρ) at experimental temperature of 313 K [25].

	P_0_/bar	M/g/mol	V_L_/10^−6^ m^3^/mol	ρ/g/cm^3^
n-Pentane	1.1509	72.15	115.2	0.626

Note that: P_0_: Saturated vapor pressure, M: Molar mass, V_L_: Molar volume, ρ: Density.

**Table 2 molecules-27-04590-t002:** Densities of the adsorbed layers and the bulk for gaseous and liquid n-pentane system, and the associated relative pressure.

System	Molecular Load	P/P_0_	n	h (nm)	ρ_al_ (kg/m^3^)	ρ_Si_ (kg/m^3^)	ρ_bulk_ (kg/m^3^)
Gas	10	0.27	1	0.32	100.79	73.91	0.89
	25	0.13	1	0.33	167.98	208.30	0.42
	50	0.46	1	0.54	470.36	295.66	1.53
	100	0.79	1	0.76	792.90	571.16	2.67
	200	0.97	2	1.05	1128.88	1122.17	3.26
Liquid	2000	--	3	1.30	1471.58	1417.83	625.71

n: number of adsorption layers, h: adsorption thickness, ρ_al_: peak density of adsorbed hydrocarbons at aluminol surface, ρ_si_: peak density of adsorbed hydrocarbons at silicate surface, ρ_bulk_: free phase density.

**Table 3 molecules-27-04590-t003:** Adsorption capacity per layer calculated directly from the simulation and predicted for given relative gas pressures on n-pentane.

System	Molecular Load	P/P_0_	C_Al_ (mg/m^2^)	C_Si_ (mg/m^2^)	C_ave_ (mg/m^2^)
gas	10	0.27	0.012	0.016	0.014
	25	0.13	0.034	0.044	0.039
	50	0.46	0.083	0.071	0.077
	100	0.79	0.18	0.12	0.15
	200	0.97	0.31	0.29	0.30
Liquid	2000	--	0.80	0.82	0.81

C_Al_: adsorption capacity per unit area of alumiol surface, C_Si_: adsorption capacity per unit area of silicate surface, C_ave_: average adsorption capacity per unit area of kaolinite.

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
