# Peer review of "Characteristics of Gaseous/Liquid Hydrocarbon Adsorption Based on Numerical Simulation and Experimental Testing"

_molecules, 2022, doi:10.3390/molecules27144590_

Round 1
Reviewer 1 Report
Thank you for inviting me to review the manuscript titled “Characteristics of Gaseous/Liquid Hydrocarbon Adsorption Based on Molecular Dynamics Simulations and Hydrocarbon Vapor Adsorption Experiments”. It represents the experimentation of HVA to calculate the adsorbed/free state oil.
1. Some sentences are general such as “is an effective method to calculate it”; “This paper focuses on solving the problem”… they should be corrected.
2. In Table 1, cm-3 should be changed to /cm3.
3. Do not repeat the words in the title to the “keywords”.
4. The last paragraph of Introduction should include the study objectives/procedures in brief.
5. The study should unveil the major gaps within the existing knowledge of HVA and estimation of the adsorbed/free state oil.
6. The results should be enriched with statistical explanations.
7. The abstract and conclusion sections should be improved to show the main research findings.
Author Response
Comments and Suggestions for Authors
Thank you for inviting me to review the manuscript titled “Characteristics of Gaseous/Liquid Hydrocarbon Adsorption Based on Molecular Dynamics Simulations and Hydrocarbon Vapor Adsorption Experiments”. It represents the experimentation of HVA to calculate the adsorbed/free state oil.
- Some sentences are general such as “is an effective method to calculate it”; “This paper focuses on solving the problem”… they should be corrected.
Thank you for your suggestion, we have changed such sentences which could be checked in the abstract.
- In Table 1, cm-3 should be changed to /cm3.
Thank you, we have revised it.
- Do not repeat the words in the title to the “keywords”.
Thank you, we have changed the title to “Characteristics of Gaseous/Liquid Hydrocarbon Adsorption Based on Numerical Simulation and Experimental Testing”.
- The last paragraph of the Introduction should include the study objectives/procedures in brief.
Thank you for your suggestion, we have reorganized the last paragraph of the introduction.
This paper seeks to evaluate the proportion of adsorbed state shale oil by HVA and MD. In section 2.1, the information of samples and the process of HVA testing were introduced. In section 2.2, the flows of MD simulation of gas/liquid hydrocarbons are shown. In section 3.1, the adsorption characteristics of gaseous and liquid hydrocarbons are compared. In section 3.2, the results of HVA experiments are interpreted at the molecular level. Finally, a correction method for HVA is proposed for the evaluation of the proportion of adsorbed state shale oil.
- The study should unveil the major gaps within the existing knowledge of HVA and estimation of the adsorbed/free state oil.
Thank you. There are still questions to be answered about HVA. Shale oil adsorption are of liquid hydrocarbons, while in the HVA, gaseous hydrocarbons are used. Currently, no relevant studies prove that the number of adsorption layers, adsorption thickness, adsorption density, and adsorption amount per unit area is directly comparable for these experiments. Therefore, the interpretation of HVA should be revisited with the aid of MD, bringing atom-level details into the process.
- The results should be enriched with statistical explanations.
Thank you. The results of hydrocarbon adsorption by HVA and the proportion of shale oil adsorbed are analyzed using statistics in Section 3
- The abstract and conclusion sections should be improved to show the main research findings.
Thank you for your suggestion, we have improved the abstract and conclusion sections which could be checked in the revised manuscript.

Reviewer 2 Report
Comments from Reviewer
Title: Characteristics of gaseous/liquid hydrocarbon adsorption based on molecular dynamics simulations and hydrocarbon vapor adsorption experiments
The current form's presentation of methods and scientific results is unsatisfactory for publication in the Molecules journal. The minor and significant drawbacks to be addressed can be specified as follows:
1. Some abbreviations are explained several times in the text. It is sufficient to do it the only first time. See, for example, (i) OSI – Lines 40 and 45 (ii) MD – Lines 42 and 60.
2. Introduction. This part of the reviewed article should be rebuilt. The last paragraph should contain clearly stated objectives.
3. Table 1. (i) References? (i) P0 – subscript. See all the manuscript.
4. Lines 132 and 133, “the aim of these tests was to obtain the adsorption isotherms of hydrocarbons and the microscopic parameters of pores in the kaolinite clay.” (i) “isotherms of hydrocarbons” or “isotherms of nitrogen”? (ii) “to obtain” or “to measure”?
5. Where is the low-temperature adsorption/desorption nitrogen isotherm? Equipment? Details of the experiment?
6. Fig. 1. (i) How were the curves in this figure calculated?
7. Lines 136 and 137, “measured by BET”. The BET is the theory method (equation). It is not the apparatus!!!
8. Line 161. factor[32] (Equations 1–2). ---> factor [32]:
9. Fig. 2. Gauges ---> gauges.
10. 2.2.5. Data Presentation and Visualisation. This paragraph is not needed. Please move the text to figure captions (Fig. 3).
11. Fig. 4. “The pink area shows octahedra, the yellow area shows tetrahedra”. I do not see these colours. Octahedron? Tetrahedron?
12. Line 287. Table 2. densities of ---> Table 2. Densities of
13. Tab. 2. 625.71 kg/cm3. How does this value relate to the experimental density?
14. Line 304, unit area. How was the surface area determined? Van der Waals one? Connolly one? Solvent accessible one?
15. Fig. 7, double-layer adsorption. Looking for a linear coefficient for two points is pointless - R2 should come out unity. Please do some additional calculations
16. Fig. 9. The authors show in Fig. a diagram of the mechanism - different gap widths. The simulations were made for one pore width. Conclusion - the authors have too little information to draw such conclusions based on their research.
17. Line 402, The characteristics of liquid hydrocarbons adsorption are multilayer adsorption. This statement is only valid for wider pores. For the narrower pores there will be an entirely different mechanism.
Sincerely,
The reviewer.
Author Response
给作者的意见和建议
审稿人的意见
标题:基于分子动力学模拟和烃蒸气吸附实验的气/液烃吸附特性
当前表格对方法和科学结果的介绍不能令人满意地在 Molecules 杂志上发表。要解决的次要和重要缺点可以指定如下:
- 有些缩略语在文中多次解释。只做第一次就足够了。例如,参见 (i) OSI – 第 40 和 45 行 (ii) MD – 第 42 和 60 行。
感谢您的建议,我们已经对整篇论文的缩写进行了检查和修改。
- This part of the reviewed article should be rebuilt. The last paragraph should contain clearly stated objectives.
Thank you for your suggestion, we have reorganized the introduction and rewritten the last paragraph.
This paper seek to evaluate the proportion of adsorbed state shale oil by HVA and MD. In section 2.1, the information of samples and the process of HVA testing were introduced. In section 2.2, the flows of MD simulation of gas/liquid hydrocarbons are shown. In section 3.1, the adsorption characteristics of gaseous and liquid hydrocarbons are compared. In section 3.2, the results of HVA experiments are interpreted at the molecular level. A correction method for HVA is proposed for the evaluation of the proportion of adsorbed state shale oil.
- Table 1. (i) References? (i) P0 – subscript. See all the manuscript.
Thank you, we have added reference in the table, and checked the subscript though the whole paper.
- Lines 132 and 133, “the aim of these tests was to obtain the adsorption isotherms of hydrocarbons and the microscopic parameters of pores in the kaolinite clay.” (i) “isotherms of hydrocarbons” or “isotherms of nitrogen”? (ii) “to obtain” or “to measure”?
Thank you for pointing out the problem. We have changed “isotherms of hydrocarbons” to “isotherms of nitrogen”, and “to obtain” to “to measure”.
- Where is the low-temperature adsorption/desorption nitrogen isotherm? Equipment? Details of the experiment?
Thank you, we have added the low-temperature adsorption/desorption nitrogen isotherm in Fig. 1a, the equipment and details of the experiment have been added in Section 2.1.1.
The kaolinite samples were milled into powder particles through 40–60 mesh (250–425 µ m) by an agate mortar. The low-temperature nitrogen adsorption/desorption ( LT-N2 A/D) were measured over relative pressures ranging from approximately 10−5 to 0.995 using an Autosorb-iQ-Station-1 instrument at 77 K to obtain the pore size distributions and specific surface areas of the kaolinite samples. Figure 1 shows the pore volume distribution determined by Barrett-Joyner-Halenda (BJH) method (with total pore volume of 0.136 cm3/g) and surface area distribution determined by BJH with a peak of 12.27 m2/g , which is in an agreement with the reported 13.1 m2/g determined by BET .
- 1. (i) How were the curves in this figure calculated?
Thank you, the curves were calculated by Barrett-Joyner-Halenda (BJH) method, which referenced the work of Bardestani.
Bardestani, R.; Patience, G. S.; Kaliaguine, S., Experimental methods in chemical engineering: specific surface area and pore size distribution measurements—BET, BJH, and DFT. The Canadian Journal of Chemical Engineering 2019, 97, (11), 2781-2791.
- Lines 136 and 137, “measured by BET”. The BET is the theory method (equation). It is not the apparatus!!!
Sorry for our carelessness. we have revised that in the paper.
- Line 161. factor[32] (Equations 1–2). ---> factor [32]:
Thank you, we have revised that in the paper.
- 2. Gauges ---> gauges.
Thank you, we have revised that word in Fig.2.
- 2.5. Data Presentation and Visualisation. This paragraph is not needed. Please move the text to figure captions (Fig. 3).
Thank you, we have moved the text to figure captions.
- 4. “The pink area shows octahedra, the yellow area shows tetrahedra”. I do not see these colours. Octahedron? Tetrahedron?
Thank you, we have added the color areas in the figure and change the words to “The pink area shows aluminol surface, the yellow area shows silicate surface”.
- Line 287. Table 2. densities of ---> Table 2. Densities of
Thank you, we have revised that.
- 2. 625.71 kg/cm3. How does this value relate to the experimental density?
Thank you, the density 625.71 kg/m3, which could be convert to 0.62571 g/cm3. And it is almost the same with experimental density 0.626 g/cm3.
- Line 304, unit area. How was the surface area determined? Van der Waals one? Connolly one? Solvent accessible one?
Thank you, we used Van der Waals one to determine the surface area. And we have added references in section 2.2.4.
- 7, double-layer adsorption. Looking for a linear coefficient for two points is pointless - R2 should come out unity. Please do some additional calculations
We are sorry for this. We will add simulations in future work. The purpose of the linear fit is to make the MD consistent with the relative pressure of the HVA for the next modeling step
- 9. The authors show in Fig. a diagram of the mechanism - different gap widths. The simulations were made for one pore width. Conclusion - the authors have too little information to draw such conclusions based on their research.
- 402行,液态烃吸附的特点是多层吸附。这个说法只对更宽的毛孔有效。对于较窄的孔,将有完全不同的机制。
谢谢你,我们完全同意你的看法。我们将在进一步的研究中增加对更窄孔隙的研究。这种狭缝孔隙设置代表了页岩储层中确定的纳米级孔隙,其中大于 8 nm 的孔隙占据了相当大的比例(71.9%)(刘,2021)。
博,L。嘉辉,S.;张,Y.;;君凌,H。小飞,F。梁,Y。吉林,X.;松辽盆地南部长岭凹陷白垩系青山口组一段页岩油储集空间及富集模式. 石油勘探与开发2021, 48, (3), 608-624。

Round 2
Reviewer 2 Report
Title: Characteristics of gaseous/liquid hydrocarbon adsorption based on experiment
Congratulations on a great job. The author has made a substantial improvement to this article. The manuscript can be accepted for publishment in the present form, however, after minor corrections.
Fig. 1. From the analysis of the adsorption values, it is seen that the unit is cm3 STP/g (or cm3/g STP). It is a typical error. The values of nitrogen adsorption for different adsorbents are typically below 1-2 cm3/g!!!! STP - Abbreviation for standard temperature (273.15 K or 0 °C) and pressure (105 Pa); usually employed in reporting gas volumes. Note that flow meters calibrated in standard gas volumes per unit time often refer to volumes at 25 °C, not 0 °C.
Sincerely,
The reviewer.
Author Response
Title: Characteristics of gaseous/liquid hydrocarbon adsorption based on experiment
Congratulations on a great job. The author has made a substantial improvement to this article. The manuscript can be accepted for publishment in the present form, however, after minor corrections.
Fig. 1. From the analysis of the adsorption values, it is seen that the unit is cm3 STP/g (or cm3/g STP). It is a typical error. The values of nitrogen adsorption for different adsorbents are typically below 1-2 cm3/g!!!! STP - Abbreviation for standard temperature (273.15 K or 0 °C) and pressure (105 Pa); usually employed in reporting gas volumes. Note that flow meters calibrated in standard gas volumes per unit time often refer to volumes at 25 °C, not 0 °C.
Thank you for your kind suggestion. We have checked the experimental data carefully, and there is no problem with the adsorption values in Fig. 1a. In addition, we compare with other researchers [1-4], the nitrogen adsorption volume near P0 is between 40~180 (cm3/g STP). So, in our opinion, the experimental results are correct.
- Caponi, N.; Collazzo, G. C.; Jahn, S. L.; Dotto, G. L.; Mazutti, M. A.; Foletto, E. L., Use of Brazilian kaolin as a potential low-cost adsorbent for the removal of malachite green from colored effluents. Materials Research 2017, 20, 14-22.
- Hezil, N.; Fellah, M.; Assala, O.; Touhami, M. Z.; Guerfi, K. In Elimination of chromium (VI) by adsorption onto natural and/or modified kaolinite, Diffusion Foundations, 2018; Trans Tech Publ: pp 106-112.
- Ghogomu, J. N.; Noufame, D. T.; Tamungang, E. B. N.; Ajifack, D. L.; Ndi, J. N.; Ketcha, J. M., Adsorption of phenol from aqueous solutions onto natural and thermallymodified kaolinitic materials. International Journal of Biological and Chemical Sciences 2014, 8, (5), 2325-2338.
- Zhang, Q.; Zhang, Y.; Chen, J.; Liu, Q., Hierarchical structure kaolinite nanospheres with remarkably enhanced adsorption properties for methylene blue. Nanoscale Research Letters 2019, 14, (1), 1-9.
